# Developmental delay and its associated factors among children under five years in urban slums of Nepal

**Anupama Bishwokarma**[1]*, **Diwash Shrestha**[2], **Kshitiz Bhujel**[3], **Natasha Chand**[4], **Laxmi Adhikari**[1], **Maheshor Kaphle**[1], **Ayurma Wagle**[1], **Isha Karmacharya**[1]

**1** Department of Public Health, CiST College, Pokhara University, Kathmandu, Nepal, **2** Department of Computer Science and Information Technology, Asian College of Higher Studies, Tribhuvan University, Kathmandu, Nepal, **3** Neuro Surgery Department, Annapurna Neurological Institute & Allied Sciences, Kathmandu, Nepal, **4** Integrated Health Information Management Section, Department of Health Services Nepal, Kathmandu, Nepal

* bishwokarma.anupama@gmail.com

**Data Availability Statement:** All relevant data are within the paper and its Supporting information files.

## Abstract

### Introduction

Children from low-resource settings are more likely to encounter those factors that adversely influence their ability to acquire developmental potential. This study was conducted to assess the developmental status and its associated factors among children under five years of slum areas of Butwal Sub Metropolitan City, Rupandehi, Nepal.

### Methods and findings

We conducted a community-based cross-sectional descriptive study using Developmental Milestone Chart (DMC) among 165 children under five years. Ethical approval was obtained from Ethical Review Board of Nepal Health Research Council. R software was used for data analysis. The association between developmental status and associated factors were examined with Chi-square and followed by logistic regression. Notably, more than half of the children (56.4%) had delayed development across two or more domains of gross motor, fine motor, language/ speech, and social development. Age, sex, socio-economic status, availability of learning materials, the occurrence of infectious diseases, and height-for-age of children were found to be significantly associated with the developmental status of children under study (p<0.05).

### Conclusions

More than half of the children taken under the study had delayed development on different four domains. Findings from the study suggest that there should be similar studies conducted among children living in slum-like conditions. Additionally, programs should be designed as such which aims to mitigate the effect of socio-economic status on child development and has learning and nutritional aspects embedded central to its deliverance.

**Funding:** The author(s) received no specific funding for this work.

**Competing interests:** The authors have declared that no competing interest exist.

**Abbreviations:** ARI, Acute Respiratory Infections; DD, Developmental delay; DMC, Developmental Milestone Chart; ECDI, Early Childhood Development Index; IWI, International Wealth Index; LMIC, Low- and middle-income countries.

## Introduction

Developmental delay (DD) in a child occurs when the child fails to achieve any one aspect of development; gross motor, fine motor, language/speech, and social development by an appropriate age [1–3]. Developmental status among children is influenced by a range of factors such as socio-economic, biological, maternal, environmental, nutritional, and genetic factors [4–7].

The children living in low-resource settings are more likely to encounter those factors that adversely influence their ability to acquire developmental potential [7–12]. In 2017, around 250 million children living in Low- and middle-income countries (LMIC) were estimated to be at risk of not achieving their full developmental potential [8]. Nearly 38% of the children living in South Asia were estimated to have low developmental scores as per Early Childhood Development Index (ECDI) [13]. The prevalence of suspected DD for ECDI for Nepal was 35.1% among children aged 36–59 months [14]. Healthcare costs associated with children who are at risk of delayed development have been found to be higher than those who are not at risk [15]. A follow-up study found that the majority of the children who were identified to be developmentally delayed at 3 years of age were either under or unemployed, living along with family and financially dependent upon their families and socially isolated [16]. If no timely identification and intervention is applied, children with delayed development living in extremely low resource settings are likely to contribute poorly school performance and subsequently generating low incomes, high fertility, and poor care for their children and eventually resulting to the intergenerational transmission of poverty [17].

According to a United Nations estimation made in 2018, 227 million of the population lived in slums or informal settlements in Southern Asia [18]. Nepal had 49.3% of the population living in slum-like conditions as of 2018 [19]. Nepal being the fastest urbanizing country in South Asia [20]. It may end up generating more slum dwellers, as informal settlements or slums have emerged to be one of the significant challenges for urban development in Nepal [21]. Furthermore, the governments in South Asia are struggling to respond to the already existing scale of growth [22].

The early childhood phase has been identified as the most effective and cost-efficient period to ensure that all children develop their full potential [23, 24]. Also, early childhood investments are claimed to have substantial benefits of preventing disease and promoting health in the long run [25]. Research estimating children at risk of development delay in overall aspects could be essential to highlight the need of interventions and policies targeting early childhood development [26]. Moreover, past studies have shown association between development outcomes and biological and anthropometrical [27], and nutritional factors [28] among children living in Nepal.

Therefore, this study was conducted with the objective to determine the prevalence of DD in four different developmental aspects such as gross motor, fine motor, language/speech, and social development and its associated factors among children under five years living in urban slum areas of Southern Nepal.

## Materials and methods

### Study design and setting

This was a community-based cross-sectional descriptive study. The data was collected from April to May 2019 among the children under 5 years of age residing in slum areas of Butwal Sub Metropolitan City. This city lies in Rupandehi district of Lumbini province situated 267 k.m. west to the capital city Kathmandu. The metropolitan city has a total population of 170,970 and 40,876 households; while it consists of slums in all its 19 wards as of 2019 [29].

## Sample size determination

The total sample size of the study was 165, calculated using Fishers' formula $n = Z^2pq/d^2$ and assuming allowable error ±0.05 at 95 percent confidence level, considering the prevalence of DD among under 5 children in an urban slum (p) 12.2% based on a previous study [2].

## Sampling procedure

Of all the 19 wards in Butwal Sub Metropolitan City, five wards with less than 50 slum households were omitted. With 14 wards remaining, 5 wards were selected randomly. Sample estimate was obtained by probability proportional to the slum household size in each ward. Further, the sampling process at each ward was initiated by identifying the center of the catchment area with the help of Google Map version 10.14.1. The household selection was done by spinning a pencil, and the first household was selected in the direction shown by the tip of the pencil. If the selected household didn't have any eligible child, the "nearest door" rule was applied; i.e., adjacent households with the nearest front door. If there were more than one eligible child in the same household, only one child was selected randomly by the lottery method.

## Ethical considerations

The study was approved by Ethical Review Board of Nepal Health Research Council. A parental written consent form was obtained before data collection. The parents were well informed about the purpose and objectives of the study, and also were notified that the participation would be voluntary.

## Study parameters

Socio-demographic variables included of age in months and sex of children as stated by respondent, family caste-ethnicity division (Dalit, Disadvantaged Janajati, Non-Dalit Disadvantaged, Terai Caste, Religious Minorities, Relatively Advantaged Janajati, Upper Caste Group) based on Health Management Information System of Nepal (HMIS) caste-ethnicity classification [30], family type (nuclear, joint and extended), and socio-economic status based on International Wealth Index (IWI) classification [31] (extremely poor, poor, middle class, upper middle class, rich). Maternal related variables were mother's age at birth, education and occupational level. Maternal educational levels included of attributes like Illiterate, Non-Formal Education, Primary Level, Lower Secondary Level, Secondary Level, Higher Secondary Level, Bachelors, Masters and Above. Mother's occupational status had following categorization as agriculture, business, private/government job, labor, homemaker and others. Environmental characteristics included availability of learning materials for children [32] (yes/no), number of languages used at home (1, 2 and >2), parental alcoholism (yes/no) and smoking (yes/no) on a daily basis. Similarly, biological variables included of birth weight (normal if ≥ 2500 gram, underweight < 2500 gram) and history of infectious diseases such as diarrhea, malaria, intestinal parasite, and others in the past six months (yes/no). However, the total number of medically reported histories of infectious diseases in the past six months by the respondents consisted of only Diarrhea, Intestinal Parasites and Acute Respiratory Infections (ARI). Height in centimeters and weight in kilograms were measured to collect anthropometric characteristics. Further, these measures were used to generate anthropometric variables such as weight for height (wasted/normal), weight for age (underweight/normal) and height for age (stunted/normal).

Later, age of the child was categorized into 5 categories as: Less than 1 year that included children from 7 months to 11 months, 1 year included of children from 12 months to 23

months, 2 years included of children from 24 months to 35 months, 3 years included children from 36 months to 47 months and 4 years included children from 48 months to 59 months. Additionally, attributes of some of the variables like family caste-ethnicity, family type, and educational level of mother and occupation of mother were modified and only two attributes were formed at the end. Attributes of family caste-ethnicity like Dalit, Disadvantaged Janajati, Non-Dalit Disadvantaged Terai Caste, and Religious Minorities were added to form Disadvantaged Group and Relatively Advantaged Janajati and Upper Caste Group were added to form Advantaged Group [33]. Similarly, Joint and extended attributes of family type were added together to form only two attributes of nuclear and Joint/Extended. Maternal educational level's attributes like Illiterate, Non-Formal Education, Primary Level, Lower Secondary Level were added to form Below Secondary level and Secondary Level, Higher Secondary Level, Bachelors, Masters and Above were added to form Secondary Level and above. Similarly, Business, Service and Labor attributes of mother's occupation were added together to form only two attributes as Working out of home and Homemaker.

## Study tools

**Assessment of developmental milestones.** DMC was used to assess a child's developmental status in two categories: Delayed and Not Delayed [34]. Gross motor, fine motor, language/speech, and social development are four domains of DMC. Children who did not meet either one or more developmental domains were considered as Developmentally Delayed. Each item in respective domains was answered either as "Yes" if the child has met the potential in a particular domain or "No" if the child has not met the potential yet. Answers for some questions were obtained directly through the mother, while for others certain activities were performed in order to check if the child has met developmental potential. Such activities would be like if the child could describe action in pictures, copies circles, etc.

DMC has been recommended for developmental screening in terms of acceptability, practicality, and implementation as part of child development monitor checkups in a low resource setting [35]. The English Version DMC tool was translated into the Nepali Language by the researcher and was again back-translated into English by a translator. DMC included for this study had 8 different developmental milestones for 8 age groups in months which are: 7–9, 10–12, 13–15, 16–18, 19–23, 24–35, 36–47, and 48–59.

Recommendations for data collection, analysis and reporting on anthropometric indicators in children under 5 years old were used for guidance [36] for taking anthropometric measurements. A Weighing machine, Stature meter and Salter scale were used to collect anthropometric data of children. Height for all children was taken in standing position. Additionally, to measure the economic situation of household and socio-economic status, we included IWI related questions within the questionnaire [31].

Pretesting of tools was done among 10% of the total sample size in the Sinamangal Slum area of Kathmandu Metropolitan City. Minor edits related to grammatical errors were done following the pretesting.

## Data collection, management, and analysis

Data collection was done using face to face interviews in the Nepali Language. Anthropometric instruments like the Weighing machine, the Stature meter, and the Salter scale were used for collecting the anthropometric data.

Epidata version 3.1 [37] was used for data entry and R Studio Version 1.1459 [38] and R language Version 3.5.1 [39] software was used for data analysis. Anthropometric data were analyzed using WHO AnthroPlus Version 3.2.2 [40]. The children who scored <-2 SD were

considered underweight (weight for age), stunted (height for age) and wasted (weight for height). Descriptive analysis, calculating frequency and percent for categorical variables and mean or median for continuous variables, was performed. Chi-square test and logistic regression were applied to determine the association between dependent and independent variables. All the statistical tests done were two-tailed and were considered statistically significant for a p-values<0.05 at 95% CI. While using logistics models, we adjusted for age of child, sex, family caste-ethnicity, family type, socio-economic status, mother's age at birth of child, educational level, occupation, books available at home, language, parental smoking, parental alcoholism, birth weight, occurrence of infectious diseases in past 6 months, weight for height, weight for age and height for age.

## Results

### Prevalence of developmental delay

In the study, more than half of the children (56.4%) had delayed development. Prevalence of DD ranged from 8.5% to 34.5% at 4 years and 1 year respectively. Of the total participants, 30.9% of them were found to have delayed development in the social development domain, followed by fine motor (28.5%) and language/speech (28.5%) and gross motor (6.7%) domain.

### Socio-demographic characteristics

The age of the children ranged from 7 months to 57 months with the mean age ± SD of 26.5 ±13.4 months. More than half (53.3%) of the children were male. Of the total participants, 66.0% and 64.8% of them belonged to disadvantaged group of family caste-ethnicity and lived in nuclear family type respectively. According to the IWI categorization, 25.5% of children belong to the upper middle class, 55.1% of the children belong to the middle class, while 19.4% of children belong to families with poor wealth index "Table 1".

**Table 1. Socio-demographic characteristics.**

| Socio-demographic characteristics | Total (n) | Percentage (%) |
|---|---|---|
| Age of child | | |
| Mean = 26.5 ± 13.4 months | | |
| ≤1 year | 27 | 16.4 |
| 1 year | 57 | 34.5 |
| 2 years | 41 | 24.8 |
| 3 years | 26 | 15.8 |
| 4 years | 14 | 8.5 |
| Sex | | |
| Female | 77 | 46.7 |
| Male | 88 | 53.3 |
| Family caste-ethnicity | | |
| Advantaged group | 56 | 34.0 |
| Disadvantaged group | 109 | 66.0 |
| Family type | | |
| Nuclear | 107 | 64.8 |
| Joint/Extended | 58 | 35.2 |
| Socio-economic status | | |
| Upper middle class | 42 | 25.5 |
| Middle class | 91 | 55.1 |
| Poor | 32 | 19.4 |

**Table 2. Maternal characteristics.**

| Maternal characteristics | Total (n) | Percentage (%) |
|---|---|---|
| Mother's age at birth of child | | |
| Mean = 24.1 ± 4.8 years | | |
| < 20 years | 40 | 24.2 |
| 20–35 years | 123 | 74.5 |
| > = 36 years | 2 | 1.2 |
| Educational level | | |
| < Secondary level | 90 | 54.5 |
| ≥ Secondary level | 75 | 45.5 |
| Occupation | | |
| Homemaker | 142 | 86.1 |
| Working out of home | 23 | 13.9 |

## Maternal characteristics

Mean age of the mother at birth of the child was 24.1±4.8 years. In terms of educational level, slightly more than half of the mothers (54.5%) were below the secondary level "Table 2".

## Environmental characteristics

Of the total 165 children, only 36.4% of the children had learning materials available at home. Children who had parents who smoked or consumed alcohol on a regular basis were 38.2% and 44.2% respectively "Table 3".

## Biological and anthropometric characteristics

The majority (86.1%) of children had normal birth weight (> = 2500 gram) with the mean birth weight (kg) 2.9 ± 0.6. Of the total 165 children, 57% of children didn't suffer from any infectious diseases (Diarrhea, Intestinal Parasites, ARI) in the past 6 months. More than 70% of the children had normal weight for height (89.1%), weight for age (86.1%), and height for age (72.7%) "Table 4".

## Factors associated with developmental delay

On bivariate analysis, the age of children (p = 0.003) and their socio-economic status (p = 0.049) were associated with their developmental status. Status of availability of learning

**Table 3. Environmental characteristics.**

| Environmental characteristics | Total (n) | Percentage (%) |
|---|---|---|
| Books available at home | | |
| No | 105 | 63.6 |
| Yes | 60 | 36.4 |
| Language | | |
| One | 121 | 73.3 |
| Two | 44 | 26.7 |
| Parental smoking | | |
| No | 102 | 61.8 |
| Yes | 63 | 38.2 |
| Parental alcoholism | | |
| No | 92 | 55.8 |
| Yes | 73 | 44.2 |

Table 4. Biological and anthropometric characteristics.

| Characteristics | Total (n) | Percentage (%) |
|---|---|---|
| **Biological characteristics** | | |
| Birth weight | | |
| Mean = 2.9±0.6 kg | | |
| Normal | 142 | 86.1 |
| Underweight | 23 | 13.9 |
| Occurrence of infectious diseases in past 6 months | | |
| No | 94 | 57.0 |
| Yes | 71 | 43.0 |
| **Anthropometric characteristics** | | |
| Weight for height | | |
| Normal | 147 | 89.1 |
| Wasted | 18 | 10.9 |
| Weight for age | | |
| Normal | 142 | 86.1 |
| Underweight | 23 | 13.9 |
| Height for age | | |
| Normal | 120 | 72.7 |
| Stunted | 45 | 27.3 |

materials for children was associated (p <0.001) with the developmental status. History of infectious diseases in the past six months was found to be associated (p = 0.027) with the developmental status of children under study. Anthropometric characteristics like height for the age of children was associated (p = 0.047) with developmental status "Table 5".

As per the results obtained, participants who were of one year of age were 3.29 times (AOR, 95% CI, 1.04–10.46) more likely to be developmentally delayed than those who were below one year of age. Female children were 0.43 times (AOR, 95% CI, 0.19–0.99) more likely to have delayed development than male children involved in the study. Children who didn't have any books available at home were 4.00 times (AOR, 95% CI, 1.31–12.26) more likely to be developmentally delayed than those who had learning materials available. Adjusted odds of being developmentally delayed was 3.79 times among children who lived with family using just one language for communication (95% CI, 1.32–10.87) compared to children who lived with family using two languages. Children who suffered from infectious diseases within the past six months of study were 2.18 times more (AOR, 95% CI, 1.01–4.69) at risk of being developmentally delayed compared to children who didn't have any occurrence of infectious disease. Likewise, the odds of being developmentally delayed was 2.07 times higher among children who were stunted (UOR, 95% CI, 1.003–4.28) than those who were normal for height for age "Table 6".

## Discussion

Our study showed a high prevalence of DD of 56.4%. Age of children, socio-economic status of family, availability of learning materials at home, the occurrence of infectious diseases in the past six months, and height for age of children were significantly associated with their developmental status.

Findings related to prevalence in our study was much higher as compared to the prevalence rate of other studies done in developed nations [41, 42]. However, in the Nepalese context the suspected DD for Nepal was reported to be 35.1% [14] as per ECDI. One of the possible

**Table 5. Association of socio-demographic, environmental, biological and anthropometric characteristics, and developmental status of children (n = 165).**

| Characteristics | Developmental status | | p-value |
|---|---|---|---|
| | Normal (n = 72) n (%) | Delayed (n = 93) n (%) | |
| **Socio-demographic characteristics** | | | |
| Age of child | | | **0.003**\*\* |
| ≤1 year | 11 (40.7) | 16 (59.3) | |
| 1 year | 14 (24.6) | 43 (75.4) | |
| 2 years | 24 (60.0) | 16 (40.0) | |
| 3 years | 15 (57.7) | 11 (42.3) | |
| 4 years | 8 (57.1) | 6 (42.9) | |
| Mean = 26.5±13.4 months | 29.3±13.5 | 24.3±13.1 | **0.016**\* |
| Sex | | | 0.284 |
| Male | 35 (39.1) | 53 (60.9) | |
| Female | 37 (48.1) | 40 (51.9) | |
| Family caste-ethnicity | | | 0.395 |
| Advantaged group | 27 (48.2) | 29 (51.8) | |
| Disadvantaged group | 45 (41.3) | 64 (58.7) | |
| Family type | | | 0.276 |
| Nuclear | 50 (46.7) | 57 (53.3) | |
| Joint/Extended | 22 (38.0) | 36 (62.0) | |
| Socio-economic status | | | **0.049**\* |
| Upper middle class | 16 (38.1) | 26 (61.9) | |
| Middle class | 47 (51.6) | 44 (48.4) | |
| Poor | 9 (28.1) | 23 (71.9) | |
| **Maternal characteristics** | | | |
| Mother's age at birth of child | | | 0.470[b] |
| <20 years | 14 (35.0) | 26 (65.0) | |
| 20–35 years | 57 (53.7) | 66 (53.7) | |
| > = 36 years | 1 (50.0) | 1 (50.0) | |
| Mean = 24.1±4.8 years | 24.8±4.9 | 23.6±4.6 | 0.112[c] |
| Educational level | | | 0.096 |
| < Secondary level | 34 (37.8) | 56 (62.2) | |
| ≥ Secondary level | 38 (50.7) | 37 (49.3) | |
| Occupation | | | 0.072 |
| Homemaker | 58 (40.8) | 84 (59.2) | |
| Working out of home | 14 (60.9) | 9 (39.1) | |
| **Environmental characteristics** | | | |
| Books available at home | | | **<0.001**\*\* |
| No | 34 (32.4) | 71 (67.6) | |
| Yes | 38 (63.3) | 22 (36.7) | |
| Language | | | 0.523 |
| One | 51 (42.1) | 70 (57.9) | |
| Two | 21 (47.7) | 23 (52.3) | |
| Parental smoking | | | 0.630 |
| No | 46 (45.1) | 56 (54.9) | |
| Yes | 26 (41.3) | 37 (58.7) | |
| Parental alcoholism | | | 0.558 |
| No | 42 (45.7) | 50 (54.3) | |
| Yes | 30 (41.1) | 43 (58.9) | |

*(Continued)*

**Table 5.** (Continued)

| Characteristics | Developmental status | | p-value |
|---|---|---|---|
| | **Normal (n = 72) n (%)** | **Delayed (n = 93) n (%)** | |
| **Biological characteristics** | | | |
| Birth weight | | | 0.639 |
| Normal | 63 (44.4) | 79 (55.6) | |
| Underweight | 9 (39.1) | 14 (60.9) | |
| Mean = 2.9±0.6 kg | 2.9±0.5 | 2.9±0.6 | 0.691$^c$ |
| Occurrence of infectious diseases in past 6 months | | | **0.027**$^*$ |
| No | 48 (51.1) | 46 (48.9) | |
| Yes | 24 (33.8) | 47 (66.2) | |
| **Anthropometric characteristics** | | | 0.151 |
| Weight for height | | | |
| Normal | 67 (45.6) | 80 (54.4) | |
| Wasted | 5 (27.8) | 13 (72.2) | |
| Weight for age | | | 0.169 |
| Normal | 65 (45.8) | 77 (54.2) | |
| Underweight | 7 (30.4) | 16 (69.6) | |
| Height for age | | | **0.047**$^*$ |
| Normal | 58 (48.3) | 62 (51.7) | |
| Stunted | 14 (31.1) | 31 (68.9) | |

p$^*$—Value significant at α <0.05, p$^{**}$—Value significant at α <0.01, p$^b$—Value from Fisher's exact test, p$^c$—Value from Independent t-test and all the rest from Chi-square test

explanations for such high concentration of DD in the present study could be that our study setting was confined to slum areas. And children living in slums are at high health risk exposure [43–45], this might lead to consequences such as delayed development. Similarly, it was found to be higher than prevalence in rural community of Rwanda (52.6%) [46], Ghana (44.6%) [47], China (35.7%) [7], India (16.2%) [48] and Malawi (11.7%) [49].

In the study, the prevalence of DD was significantly higher among 1-year children as compared to other age groups. A study conducted in a similar study setting using the same assessment tool showed similar results of having DD (20.3%) at 12–23 months of age [2]. Further among the socio-demographic variables, association was obtained between the socio-economic status of the family they belonged to and their developmental status. In a study done in China [50] and Iran [1] also found similar results. Likewise, an estimate made in 2017 indicated that children in low and middle-income countries are at risk of not achieving their full developmental potential [8]. This may be due to the relative effect of financial instability [51] on variables such as birth weight, nutritional intake, inter-parental and parent/child interactions, etc., which in turn is known to be affecting the range of child developmental outcomes [52, 53].

Maternal education is an important determinant for child health [54, 55] as it has a positive effect on child health through an increased probability of; use of prenatal care [56], child health service utilization [57], being more receptive to modern medical treatments [58]. However, in the current study there was no significant effect of maternal education on developmental status of children.

Availability of any form of learning materials for children reduces the risk of increasing delayed development among children [50], particularly speech and language skills [48].

**Table 6. Factors associated with developmental status of children (n = 165).**

| Characteristics | Developmental status | |
|---|---|---|
| | **Unadjusted** | **Adjusted** |
| | **OR (95% CI)** | **OR (95% CI)** |
| **Socio-demographic characteristics** | | |
| Age of child | | |
| ≤ 1 year | Ref. | Ref. |
| 1 year | 2.11 (0.79–5.60) | **3.29 (1.04–10.46)**\* |
| 2 years | 0.49 (0.18–1.31) | 0.58 (0.17–1.99) |
| 3 years | 0.50 (0.17–1.50) | 2.01 (0.42–9.49) |
| 4 years | 0.52 (0.14–1.91) | 1.36 (0.25–7.39) |
| Sex | | |
| Male | Ref. | Ref. |
| Female | 0.71 (0.38–1.32) | **0.43 (0.19–0.99)**\* |
| Family caste-ethnicity | | |
| Advantaged group | Ref. | Ref. |
| Disadvantaged group | 1.32 (0.69–2.53) | 1.75 (0.71–4.34) |
| Family Type | | |
| Nuclear | Ref. | Ref. |
| Joint / Extended | 1.44 (0.75–2.76) | 1.90 (0.80–4.53) |
| Socio-economic Status | | |
| Upper middle class | Ref. | Ref. |
| Middle class | 0.58 (0.27–1.22) | 0.93 (0.36–2.40) |
| Poor | 1.57 (0.58–4.24) | 1.38 (0.37–5.17) |
| **Maternal characteristics** | | |
| Mother's age at birth of child | | |
| <20 years | 1.46 (0.08–25.81) | 0.13 (0.003–5.34) |
| 20–35 years | 1.27 (0.08–20.67) | 0.26 (0.01–9.44) |
| > = 36 years | Ref. | Ref. |
| Educational Level | | |
| < Secondary Level | 1.69 (0.91–3.15) | 2.11 (0.96–4.66) |
| ≥Secondary Level | Ref. | Ref. |
| Occupation | | |
| Homemaker | Ref. | Ref. |
| Working out of Home | 0.62 (0.31–1.22) | 0.60 (0.26–1.41) |
| **Environmental characteristics** | | |
| Books available at home | | |
| Yes | Ref. | Ref. |
| No | **3.61 (1.85–7.02)**\*\* | **4.00 (1.31–12.26)**\* |
| Language | | |
| One | 1.25 (0.63–2.51) | **3.79 (1.32–10.87)**\* |
| Two | Ref. | Ref. |
| Parental Smoking | | |
| No | Ref. | Ref. |
| Yes | 1.17 (0.62–2.21) | 1.15 (0.44–2.99) |
| Parental Alcoholism | | |
| No | Ref. | Ref. |
| Yes | 1.20 (0.65–2.24) | 0.94 (0.36–2.50) |

(*Continued*)

**Table 6.** (Continued)

| Characteristics | Developmental status | |
|---|---|---|
| | Unadjusted | Adjusted |
| | OR (95% CI) | OR (95% CI) |
| **Biological characteristics** | | |
| Birth Weight | | |
| Normal | Ref. | Ref. |
| Underweight | 1.24 (0.50–3.05) | 1.68 (0.55–5.14) |
| Occurrence of Infectious Diseases in past 6 months | | |
| No | Ref. | Ref. |
| Yes | **2.04 (1.08–3.86)\*** | **2.18 (1.01–4.69)\*** |
| **Anthropometric characteristics** | | |
| Weight for height | | |
| Normal | Ref. | Ref. |
| Wasted | 2.18 (0.74–6.42) | 2.26 (0.32–15.75) |
| Weight for age | | |
| Normal | Ref. | Ref. |
| Underweight | 1.93 (0.75–4.98) | 1.91 (0.30–12.35) |
| Height for age | | |
| Normal | Ref. | Ref. |
| Stunted | **2.07 (1.003–4.28)\*** | 1.55 (0.61–3.94) |

p\*—Value significant at $\alpha < 0.05$, p\*\*—Value significant at $\alpha < 0.001$, All the covariates are adjusted.

Relevant findings have been obtained in studies in the past supporting this statement [7, 59, 60]. Similarly, our study also revealed that not having any form of learning materials at home for children increased the likelihood of being developmentally delayed in children below five years.

Though many studies have shown there's a strong link between the low birth weight of children with their developmental status [61, 62]. However, children in our study had no consequence of being low weight at birth to their development, which is similar to the findings from the study done among Chinese [7] and Brazilian children [63]. One possible explanation would be the increased reach of obstetric and neonatal care to those children which might have reduced the disadvantages of being born with low weight [64]. However, we lack evidence to support the improved obstetric and neonatal care service provision and service utilization in the current study area [65]. While the children living in low socioeconomic status have high chances of occurrence of infectious diseases given the poor sanitation conditions [1], the occurrence of infectious disease in early years of life can lead to delayed development [66, 67]. We found a similar relation of delayed development among those who suffered infectious disease in past six-months prior to the data collection.

Stunting in early childhood particularly is associated with low cognitive skills, thus affecting developmental status [8, 68, 69]. A study done in rural areas of India showed that malnourished children attained developmental milestones at later age [68]. Similarly, the findings from a study done in LMICs shows that the children are at high-risk of not achieving developmental potential due to stunting [8]. Likewise, our study also indicated that stunting at early years of life is related to increase the odds of being developmentally delayed.

Our study is one of those minimal studies that presents the developmental status of children living in urban slums in Nepal; one of the dimmed areas of developmental aspect for children

from low-resource settings. The tool we used for the study assess the developmental status of a child in four developmental aspects such as gross motor, fine motor, language/ speech, and social development which are one of the major aspects considered primary for any of the developmental tools. However, we had few limitations such as; not having neighborhood and paternal characteristics incorporated within the questionnaire, having relatively small sample size. Also, there could have been a recall bias at times as there were few questions that would require respondents to report; if their child performed any of the activities, weight at birth and occurrence of infectious diseases within the past 6 months. Additionally, use of a cross-sectional study method limits the potential to examine the causal relationship.

## Conclusions

Our study found that more than half of the children were found to be developmentally delayed in the study area. Age, socio-economic status, availability of learning materials, occurrence of infectious disease and height for age of children were found to be significantly associated with developmental status of children under study. Findings from the study suggest that investigations need to focus on overall developmental aspects of early childhood development of children. Additionally, programs should be designed as such which aims to mitigate the effect of SES on child development and has learning and nutritional aspects embedded central to its deliverance.

## Supporting information

**S1 File.**
(XLSX)

## Acknowledgments

Authors would like to express sincere acknowledgement to Central Institute of Science and Technology College and Butwal Sub-metropolitan City Office for giving permission to conduct research, and Nepal Health Research Council for ethical approval. Authors are grateful to all the parents who gave consent to involve their children and most importantly acknowledge all the children involved in the study.

## Author Contributions

**Conceptualization:** Anupama Bishwokarma.

**Data curation:** Diwash Shrestha.

**Formal analysis:** Anupama Bishwokarma, Diwash Shrestha, Kshitiz Bhujel.

**Investigation:** Anupama Bishwokarma.

**Methodology:** Anupama Bishwokarma, Isha Karmacharya.

**Project administration:** Anupama Bishwokarma, Isha Karmacharya.

**Supervision:** Isha Karmacharya.

**Visualization:** Diwash Shrestha.

**Writing – original draft:** Anupama Bishwokarma, Diwash Shrestha, Kshitiz Bhujel, Natasha Chand.

**Writing – review & editing:** Anupama Bishwokarma, Diwash Shrestha, Kshitiz Bhujel, Natasha Chand, Laxmi Adhikari, Maheshor Kaphle, Ayurma Wagle, Isha Karmacharya.

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
