## [Decision Letter · Decision Letter 0]

28 Apr 2021

PONE-D-21-05500

Developmental delay and its associated factors among children under five years in urban slums of Nepal

PLOS ONE

Dear Dr. Karmacharya,

Thank you for submitting your manuscript to PLOS ONE. After careful consideration, we feel that it has merit but does not fully meet PLOS ONE’s publication criteria as it currently stands. Therefore, we invite you to submit a revised version of the manuscript that addresses the points raised during the review process.

 I would like to see a revised version of this paper. Thus, I am going with a decision of revise and resubmit according to your suggestions.

We look forward to receiving your revised manuscript.

Kind regards,

Srinivas Goli, Ph.D.

Academic Editor

PLOS ONE

Additional Editor Comments:

Based on reviewers suggestion and my own reading, I am going with a decision of Major revision for this paper. Looking forward to see the revised version of the paper.

Journal Requirements:

2.  In the Methods section, please clearly state whether IRB approval was obtained from the Nepal Health Research Council and please  amend your current ethics statement to include the full name of the ethics committee/institutional review board(s) that approved your specific study.

3. When reporting the results of qualitative research, we suggest consulting the COREQ guidelines: http://intqhc.oxfordjournals.org/content/19/6/349. In this case, please consider including more information on the number of interviewers, their training and characteristics.

Please provide additional details regarding parent consent. In the ethics statement in the Methods and online submission information, please ensure that you have specified (1) whether consent was suitably informed and (2) what type you obtained (for instance, written or verbal).

Please provide further clarification on the study variables which were obtained through 'certain activities which were performed in order to check if the check has met developmental potential'. In particular please state which were under taken, the training and background of the investigator under taking these measurements.

Reviewers' comments:

Reviewer's Responses to Questions

**Comments to the Author**

1. Is the manuscript technically sound, and do the data support the conclusions?

Reviewer #1: Partly

Reviewer #2: Partly

Reviewer #3: Yes

2. Has the statistical analysis been performed appropriately and rigorously? 

Reviewer #1: Yes

Reviewer #2: No

Reviewer #3: Yes

3. Have the authors made all data underlying the findings in their manuscript fully available?

Reviewer #1: Yes

Reviewer #2: No

Reviewer #3: Yes

4. Is the manuscript presented in an intelligible fashion and written in standard English?

Reviewer #1: No

Reviewer #2: No

Reviewer #3: Yes

5. Review Comments to the Author

Reviewer #1: Overview: This manuscript is based on primary study. The study is on important topic and is very interesting. The authors tried to include important part in the introduction however, significant work need to be done in the literature gap and significance of study. Additionally, it is advised to add peer-reviewed scientific studies and national reports related to child developmental delay in the introduction part to provide a clear picture. The method section would benefit from mentioning reason for purposive sampling, validity of tools in Nepalese context. Further, adding definition on independent variables with refer to appropriate sources is needed. Response rate in the result section is missing. The discussion part can be improved by providing more possible explanation of the casual relationship of developmental delay and other factors. Authors haven’t mention about strengths and weakness of this study. The recommendations are not clearly stated.

Abstract: In the conclusion, there is new information which should be included in the result section.

Introduction: This section could benefit from careful clarification. It has not been written and developed logically; paragraphs are not connected logically. Further, justification for this research is not extensive. Authors are advised to include previous studies on a similar topic in the Nepalese context if applicable. There is no clear indication of a literature gap in the introduction section. The manuscript would be more comprehensive if the authors add on that part.

Methods: This section could benefit from careful clarification

Study design and setting: This section needs careful English edit.

Sampling procedure: Authors are advised to include the rationale of doing purposive sampling.

Study tools: Please mention the validity and reliability of tools. Elaboration needed on this part “certain activities were performed in order to check if the child has met developmental potential.” Authors are advised to include definition of independent variables. How did authors define family ethnicity, socio-economic status, education status of mother, occupation, availability of learning materials, mother’s age, alcohol intake, smoking, history of infectious diseases, height, and weight? The definition should be supported with references. By doing so, the study looks more scientific and credible.

Result: Authors are advised to include response rate. If there is missing data, how did you deal with it? The study result is influenced by missing data.

Discussion: This section makes interesting points; however, I think the authors can organise it a bit better by adding more explanations and relevant references after explanation.

Maternal related characteristics: This section needs English edit. There is lack of coherence in writing. Authors are advised to elaborate further on how maternal education influences children development.

Environment related characteristics: Please provide credible source on this statement, ” The study revealed that not having any form of learning materials at home for children increased the likelihood of being developmentally delayed in children below five years by four-folds.”

Biological and anthropometric characteristics: Please add reference to this, “This might enhance the chances of exposure of children to factors for developing infectious diseases, thus affecting their developmental potential”.

Please elaborate more on this statement, “Present study shows that stunted children were not able to achieve developmental 261 scores as compared to nourished children whose results are in alignment to study 262 done in LMICs”.

Please include strength and limitations of this study.

Conclusion and recommendations: Conclusions need careful revision. Recommendation part is not clear. Authors are advised to include programmatic recommendations and meaningful suggestions for future research in the Child developmental delay.

Thank you for the opportunity to review your work.

Reviewer #2: The introduction is not sound and complete to explain the parameters of developmental delay and slum like areas.

There is not enough information on what were the parameters which were observed to assess the developmental delay.

It will good to have a reference for the standard of categories like education, family ethnicity, socioeconomic status and occupation.

It will also be great to have a brief description on how the anthropometric measurement were made.

The scale used to assess the developmental delay needs more explanation and clarification.

Discussion needs to be explanatory and contextual.

When i downloaded the article all the lines were numbered, major corrections need to be done with the formatting, alignment of the text.

Reviewer #3: Dear Authors,

Your study to determine the prevalence of development delay in the urban slum community of Nepal is really interesting which could guide policymakers to design programs in those community.

My comments:

1. Most of the time, you used passive voice while explaining the results and the things that you performed. Please try to use active voice in the sentence which makes the piece direct, clear and concise.

For example: line 36-38, you can tailor in this way.

eg. We conducted a community based cross sectional descriptive study using Developmental Milestone Chart (DMC) among 165 children of under five years.

2. Line 62-64

Please revise this sentence. "However., children in slums .......developmental potential." You provided the data of prevalence of DD in the world and South Asia. Following that you started with 'however' showing the risk of children in slum. Do try to make a good connection with these two sentences.

Further in the background section, please provide some details about slum community in Nepal.

3. Line 64-65

What are the handful data? Can you provide some specific examples with references?

4. Line 97-100

Please provide reference of DMC.

5. Line 138-139

provide reference for this softwares.

6. Line 154-155

Please don't make the sentence redundant. Instead of saying "The study population included a majority of male children

155 (53.3%) than female children (46.7%)." You can simply write: "More than half (53.3%) of the children were male."

7. Line 164

Were there mothers in your study who had no formal education? You categorized mother education into two categories only. Could you provide more details?

8. Line 223-224

Revise the sentence

Suggestion: A study conducted in urban slum of India using same assessment tool showed similar result; ......% of under one children had developmental delay.

9. Line 230-233

Have you explored any neighborhood characteristics in the literature that has effect on developmental delay in children?

I believe there are several studies on this. You can discuss this aspect as well.

10. Line 234 to 241

What could be the possible explanation of no statistically significant association of maternal characteristics with DD?

11. Line 251-253

What could be the possible explanation of this findings? You need to discuss it here.

12. Line 257-259

What is the pathway that infectious disease enhances DD? What are the infectious diseases you asked for?

13. Above line 268

What are the limitations of your study? This section is missing.

- small sample size, purposive sampling

- missing independent variables such as neighborhood effect and so on..

In summary, you need to revise your writing using active voice, especially when you are talking about your work. And more importantly, you need to discuss more in your discussion section.

Best wishes.

Thank you.

6. PLOS authors have the option to publish the peer review history of their article (what does this mean?). If published, this will include your full peer review and any attached files.

Reviewer #1: **Yes: **Mandira Adhikari

Reviewer #2: No

Reviewer #3: No

---

## [Author Response · Author response to Decision Letter 0]

1 Oct 2021

We are grateful to all the reviewers for their time and effort to read and comment on our paper. Their comments are constructive and the paper has benefitted from it. Below we present the point-by-point response to the reviewer’s comments. 

Additional Editor Comments:

Based on reviewers suggestion and my own reading, I am going with a decision of Major revision for this paper. Looking forward to see the revised version of the paper.

Thank you so much for your considerations. We have uploaded the revised version of the paper as well.

Journal Requirements:

 We have adhered the manuscript to above mentioned style templates.

2. In the Methods section, please clearly state whether IRB approval was obtained from the Nepal Health Research Council and please amend your current ethics statement to include the full name of the ethics committee/institutional review board(s) that approved your specific study.

Thank you for your feedback. We have added the name of ethic committee that approved our study as following.

In Methods and Materials,

The study was approved by Ethical Review Board at Nepal Health Research Council.

3. When reporting the results of qualitative research, we suggest consulting the COREQ guidelines: http://intqhc.oxfordjournals.org/content/19/6/349. In this case, please consider including more information on the number of interviewers, their training and characteristics.

Please provide additional details regarding parent consent. In the ethics statement in the Methods and online submission information, please ensure that you have specified (1) whether consent was suitably informed and (2) what type you obtained (for instance, written or verbal).

We have mentioned about the parental consent as such:

In Methods and Materials,

A parental written consent form was obtained before data collection.

Please provide further clarification on the study variables which were obtained through 'certain activities which were performed in order to check if the check has met developmental potential'. In particular please state which were under taken, the training and background of the investigator under taking these measurements.

Thank you for your suggestions. We have provided clarification on the study variables in the manuscript. The investigator involved in the have a Public Health background and they followed the guidance provided by the experts through out the period of data collection. Data was collected only after in-detailed consultation with co-investigators who previously used the tool in similar study setting. 

We will not restrict data and have added the data as Supporting Information files.

 We have added data set as Supporting Information files.

Reviewers' comments:

Reviewer's Responses to Questions

Comments to the Author

1. Is the manuscript technically sound, and do the data support the conclusions?

Reviewer #1: Partly

Reviewer #2: Partly

Reviewer #3: Yes

We have worked to make the manuscript technically sound and revised the conclusions based on the findings.

2. Has the statistical analysis been performed appropriately and rigorously?

Reviewer #1: Yes

Reviewer #2: No

Reviewer #3: Yes

Thank you for your feedback. We have performed statistical analysis under proper guidance from expertise. 

3. Have the authors made all data underlying the findings in their manuscript fully available?

Reviewer #1: Yes

Reviewer #2: No

Reviewer #3: Yes

We will not restrict data and have added the data as Supporting Information files.

4. Is the manuscript presented in an intelligible fashion and written in standard English?

Reviewer #1: No

Reviewer #2: No

Reviewer #3: Yes

Thank you for your suggestions. We have revised entire manuscript in terms of the writing.

Reviewer Comments:

Reviewer 1

1. Overview: This manuscript is based on primary study. The study is on important topic and is very interesting. The authors tried to include important part in the introduction however, significant work need to be done in the literature gap and significance of study. Additionally, it is advised to add peer-reviewed scientific studies and national reports related to child developmental delay in the introduction part to provide a clear picture. 

Thank you for your feedback. We have added articles that reflect current gaps.

2. The method section would benefit from mentioning reason for purposive sampling, validity of tools in Nepalese context.

Thank you so much for pointing it out. We would like to apologize about the statement where we mentioned that we used purposive sampling, as there has been a mistake from our side to provide the information related to selecting each ward of the study area. We had selected the wards of the study setting using the lottery method. To back this with evidence we would like to mention that it was well documented in our submitted proposal for Ethical Approval to Nepal Health Research Council (NHRC) and e-poster presentation done at the NHRC Summit of 2021 (link to the e-poster: Summit NHRC – Seventh National Summit). Now, about the validity of tools in Nepalese context, this tool has not been yet validated in the Nepalese context, but has been used in a similar low-resource setting. 

3. Further, adding definition on independent variables with refer to appropriate sources is needed. 

Thank you for your feedback. We have added operational definitions that were considered for this study for above mentioned independent variables and cited them respectively wherever appropriate.

4. Response rate in the result section is missing.

All the individuals who were reached approved the study as participants, therefore, we have not included non-response rate in the study.

5. The discussion part can be improved by providing more possible explanation of the casual relationship of developmental delay and other factors.

We have revised the discussion section including more of the causal relationship between developmental delay and other factors, and cited them accordingly.

6. Authors haven’t mention about strengths and weakness of this study.

We have mentioned strengths and weaknesses of the study. 

In Discussion section:

Our study is one of those minimal studies that presents the developmental status of children living in urban slums in Nepal; one of the dimmed areas of developmental aspect for children from low-resource settings. The tool we used for the study assess the developmental status of a child in four developmental aspects such as gross motor, fine motor, language/ speech, and social development which are one of the major aspects considered primary for any of the developmental tools. However, we had few limitations such as; not having neighborhood and paternal characteristics incorporated within the questionnaire, having relatively small sample size. Also, there could have been a recall bias at times as there were few questions that would require respondents to report; if their child performed any of the activities, weight at birth and occurrence of infectious diseases within the past 6 months. Additionally, use of a cross-sectional study method limits the potential to examine the causal relationship.

7. The recommendations are not clearly stated.

We have revised the recommendations.

In Conclusion section:

Our study found that more than half of the children were found to be developmentally delayed in the study area. Age, socio-economic status, availability of learning materials, occurrence of infectious disease and height for age of children were found to be significantly associated with developmental status of children under study. Findings from the study suggest that there should be similar studies conducted among children living in slum-like conditions. Such investigations should focus on overall developmental aspects of ECD of children. Additionally, programs should be designed as such which aims to mitigate the effect of SES on child development and has learning and nutritional aspects embedded central to its deliverance.

8. Abstract: In the conclusion, there is new information which should be included in the result section.

We appreciate your feedback; however, we could not be certain about the specific new information you mentioned. Therefore, we have added a statement in the abstract which was in Conclusions previously considering it might be the new information you pointed out. Please kindly consider reviewing it once.

In Abstract:

Notably, more than half of the children (56.4%) had delayed development across two or more domains of gross motor, fine motor, language/ speech, and social development. 

9. Introduction: This section could benefit from careful clarification. It has not been written and developed logically; paragraphs are not connected logically. Further, justification for this research is not extensive. Authors are advised to include previous studies on a similar topic in the Nepalese context if applicable. There is no clear indication of a literature gap in the introduction section. The manuscript would be more comprehensive if the authors add on that part.

Thank you for your suggestions. We have worked on revising Introduction as per the comments and suggestions received.

10. Methods: This section could benefit from careful clarification.

Thank you for your feedback. We have worked on revising the methods section.

11. Study design and setting: This section needs careful English edit.

We have worked on grammatical errors and language used here.

12. Sampling procedure: Authors are advised to include the rationale of doing purposive sampling.

Thank you so much for pointing it out. We would like to apologize about the statement where we mentioned that we used purposive sampling, as there has been a mistake from our side to provide the information related to selecting each ward of the study area. We had selected the wards of the study setting using the lottery method. To back this with evidence we would like to mention that it was well documented in our submitted proposal for Ethical Approval to Nepal Health Research Council (NHRC) and e-poster presentation done at the NHRC Summit of 2021 (link to the e-poster: Summit NHRC – Seventh National Summit). 

13. Study tools: Please mention the validity and reliability of tools. 

Thank you for your feedback. The validity of tools, this tool has not been yet validated in the Nepalese context, but has been recommended to be used in a low-resource setting. 

14. Elaboration needed on this part “certain activities were performed in order to check if the child has met developmental potential.”

We have added the following statement and provided the details of the activities performed as Supporting Information.

In Materials and Methods section:

Answers for some questions were obtained directly through the mother, while for others certain activities were performed in order to check if the child has met developmental potential. Such activities would be like if the child could describe action in pictures, copies circles, etc.

15. Authors are advised to include definition of independent variables.

Thank you for your feedback. As per the suggestion we have added operational definitions formed for independent variables used in this study whenever applicable. 

16. How did authors define family ethnicity, socio-economic status, education status of mother, occupation, availability of learning materials, mother’s age, alcohol intake, smoking, history of infectious diseases, height, and weight? The definition should be supported with references. By doing so, the study looks more scientific and credible. 

Thank you for your feedback. We have added operational definitions that were considered for this study for above mentioned independent variables and cited them respectively whenever applicable.

17. Result: Authors are advised to include response rate. If there is missing data, how did you deal with it? The study result is influenced by missing data.

All the individuals who were reached approved to participate in the study. Therefore, we have not included non-response rate in the study. Fortunately, there was no missing data that could have influenced our study results.

18. Discussion: This section makes interesting points; however, I think the authors can organise it a bit better by adding more explanations and relevant references after explanation.

We have revised the discussion section and cited them with relevant references.

19. Maternal related characteristics: This section needs English edit. There is lack of coherence in writing. 

We have worked on grammatical errors and language here.

20. Maternal related characteristics: Authors are advised to elaborate further on how maternal education influences children development.

We have provided elaboration on how maternal education influences children development with relevant references.

In Discussion:

Maternal education is an important determinant for child health [51,52] as it has a positive effect on child health through an increased probability of; use of prenatal care [53], child health service utilization [54], being more receptive to modern medical treatments [55]. However, in the current study there was no significant effect of maternal education on developmental status of children. 

21. Environment related characteristics: Please provide credible source on this statement, ” The study revealed that not having any form of learning materials at home for children increased the likelihood of being developmentally delayed in children below five years by four-folds.”

This statement wanted to indicate the findings of our own study. We have revised the statement in environmental related characteristics. 

In Discussion:

Similarly, our study also revealed that not having any form of learning materials at home for children increased the likelihood of being developmentally delayed in children below five years. 

22. Biological and anthropometric characteristics: Please add reference to this, “This might enhance the chances of exposure of children to factors for developing infectious diseases, thus affecting their developmental potential”.

We have revised the above statement in biological and anthropometric characteristics and cited accordingly.

In Discussion: 

While the children living in low socioeconomic status have high chances of occurrence of infectious diseases given the poor sanitation conditions [1], the occurrence of infectious disease in early years of life can lead to delayed development [63,64]. 

23. Please elaborate more on this statement, “Present study shows that stunted children were not able to achieve developmental 261 scores as compared to nourished children whose results are in alignment to study 262 done in LMICs”.

We have revised the above statement. 

24. Please include strength and limitations of this study.

We have mentioned strengths and weaknesses of the study. 

In Discussion section:

Our study is one of those minimal studies that presents the developmental status of children living in urban slums in Nepal; one of the dimmed areas of developmental aspect for children from low-resource settings. The tool we used for the study assess the developmental status of a child in four developmental aspects such as gross motor, fine motor, language/ speech, and social development which are one of the major aspects considered primary for any of the developmental tools. However, we had few limitations such as; not having neighborhood and paternal characteristics incorporated within the questionnaire, having relatively small sample size. Also, there could have been a recall bias at times as there were few questions that would require respondents to report; if their child performed any of the activities, weight at birth and occurrence of infectious diseases within the past 6 months. Additionally, use of a cross-sectional study method limits the potential to examine the causal relationship.

25. Conclusion and recommendations: Conclusions need careful revision. Recommendation part is not clear. Authors are advised to include programmatic recommendations and meaningful suggestions for future research in the Child developmental delay.

Thank you for your feedback. We have revised conclusions and recommendation as following:

In Conclusions section:

Our study found that more than half of the children were found to be developmentally delayed in the study area. Age, socio-economic status, availability of learning materials, occurrence of infectious disease and height for age of children were found to be significantly associated with developmental status of children under study. Findings from the study suggest that there should be similar studies conducted among children living in slum-like conditions. Such investigations should focus on overall developmental aspects of ECD of children. Additionally, programs should be designed as such which aims to mitigate the effect of SES on child development and has learning and nutritional aspects embedded central to its deliverance.

Reviewer 2

1. The introduction is not sound and complete to explain the parameters of developmental delay and slum like areas.

Thank you for your feedback. We have worked on improvising language.

2. There is not enough information on what were the parameters which were observed to assess the developmental delay.

Thank you for your feedback. Developmental delay in this study was assessed for four different domains such as Gross motor, fine motor, language/speech, and social development We have provided further information about these parameters with relevant references. 

3. It will good to have a reference for the standard of categories like education, family ethnicity, socioeconomic status and occupation.

We have classified and added relevant references whenever possible for education, family ethnicity, socioeconomic status and occupation accordingly. 

4. It will also be great to have a brief description on how the anthropometric measurement were made.

We have provided further description on how the anthropometric measurements were made. 

In Materials and methods section:

Recommendations for data collection, analysis and reporting on anthropometric indicators in children under 5 years old were used for guidance [33] for taking anthropometric measurements. A Weighing machine, Stature meter and Salter scale were used to collect anthropometric data of children. Height for all children was taken in standing position. Additionally, to measure the economic situation of household and socio-economic status, we included IWI related questions within the questionnaire.

5. The scale used to assess the developmental delay needs more explanation and clarification.

We have added the following statement to address the comment by providing further explanation and clarification about the tool used to assess the developmental delay.

6. Discussion needs to be explanatory and contextual.

Thank you for your valuable feedback. We have worked on making discussion more explanatory and contextual.

7. When i downloaded the article all the lines were numbered, major corrections need to be done with the formatting, alignment of the text.

Thank you for pointing this out. Manuscript includes line numbers, as per the guidelines for submission. This might be the reason for the occurrence of the above-mentioned condition. We will ensure that this problem will not happen next time.

Reviewer 3

Your study to determine the prevalence of development delay in the urban slum community of Nepal is really interesting which could guide policymakers to design programs in those community.

Thank you for your motivating words.

1. Most of the time, you used passive voice while explaining the results and the things that you performed. Please try to use active voice in the sentence which makes the piece direct, clear and concise. For example: line 36-38, you can tailor in this way. eg. We conducted a community based cross sectional descriptive study using Developmental Milestone Chart (DMC) among 165 children of under five years.

Thank you for your feedback. We have worked on revising sentences into active voice. And added the above statement in the abstract section. 

2. Line 62-64: Please revise this sentence. "However., children in slums .......developmental potential." You provided the data of prevalence of DD in the world and South Asia. Following that you started with 'however' showing the risk of children in slum. Do try to make a good connection with these two sentences.

Thank you for pointing it out. We have acknowledged the suggestion and have tried to establish connection between sentences with revision of the above statement in the Introduction section.

3. Further in the background section, please provide some details about slum community in Nepal.

We have added information as follows:

In Introduction section:

According to a United Nations estimation made in 2018, 227 million of the population lived in slums or informal settlements in Southern Asia [18]. Nepal had 49.3% of the population living in slum-like conditions as of 2018 [19]. Nepal being the fastest urbanizing country in South Asia [20]. It may end up generating more slum dwellers, as informal settlements or slums have emerged as one of the significant challenges to urban development in Nepal [21]. Furthermore, the governments in South Asia are struggling to respond to the already existing scale of growth [22]. 

4. Line 64-65: What are the handful data? Can you provide some specific examples with references?

We have revised the statement and added citations accordingly.

In Introduction section:

However, only few studies have been conducted in developing countries like Nepal, that provide evidence for estimated overall child development and associated factors among children living in urban slums in Nepal.

5. Line 97-100 Please provide reference of DMC.

Thank you for your feedback. Reference to the tool (DMC) was provided accordingly.

In Materials and methods section:

DMC was used to assess a child’s developmental status in two categories: Delayed and Not Delayed [31]. Gross motor, fine motor, language/speech, and social development are four domains of DMC. 

6. Line 138-139, provide reference for this softwares.

Thank you for your feedback. Reference to the softwares (Epidata 3.1, R Studio 1.1.459, R language version 3.5.1 and WHO AnthroPlus Version 3.2.2) was provided accordingly.

In Materials and methods section:

Epidata version 3.1 [34] was used for data entry and R Studio Version 1.1.459 [35] and R language Version 3.5.1 [36] software was used for data analysis. Anthropometric data were analyzed using WHO AnthroPlus Version 3.2.2 [37]. 

7. Line 154-155 Please don't make the sentence redundant. Instead of saying "The study population included a majority of male children 155 (53.3%) than female children (46.7%)." You can simply write: "More than half (53.3%) of the children were male."

Thank you for your suggestion. We have added the provided statement accordingly.

In Results section:

The age of the children ranged from 7 months to 57 months with the mean age ± SD of 26.5±13.4 months. More than half (53.3%) of the children were male. Of the total participants, 66.0% and 64.8% of them belonged to underprivileged family ethnicity and lived in nuclear family type respectively.

8. Line 164 Were there mothers in your study who had no formal education? You categorized mother education into two categories only. Could you provide more details?

We attributed maternal educational levels into Illiterate, Non-Formal Education, Primary Level, Lower Secondary Level, which were later added to form < Secondary level and Secondary Level Higher, Secondary Level, Bachelors, Masters and Above were added to form ≥ Secondary level. 

9. Line 223-224, Revise the sentence, Suggestion: A study conducted in urban slum of India using same assessment tool showed similar result; ......% of under one children had developmental delay.

Thank you for your suggestions. We have added the provided statement accordingly. 

In the study, the prevalence of DD was significantly higher among 1-year children as compared to other age groups. A study conducted in a similar study setting using the same assessment tool showed similar results of having developmental delay (20.3%) at 12-23 months of age [2]. 

10. Line 230-233, Have you explored any neighborhood characteristics in the literature that has effect on developmental delay in children?I believe there are several studies on this. You can discuss this aspect as well.

We agree with the comment. However, we have not discussed any of the neighborhood characteristics in the study. We have added this in the limitation of study. 

11. Line 234 to 241, What could be the possible explanation of no statistically significant association of maternal characteristics with DD?

Thank you for your feedback. However, we could not come up with a possible explanation of no statistically significant association of maternal characteristics with developmental delay of children in this study.

12. Line 251-253, What could be the possible explanation of this findings? You need to discuss it here.

We have provided a possible explanation for this finding as such.

In Discussion section:

Though many studies have shown there’s a strong link between the low birth weight of children with their developmental status [58,59]. However, children in our study didn’t have the disadvantage of being born with low birth weight, which is consistent with study done among Chinese [7] and Brazilian children [60]. One possible explanation would be the increased reach of obstetric and neonatal care to those children which might have reduced the disadvantages of being born with low weight [61]. However, we lack evidence to support the improved obstetric and neonatal care service provision and service utilization in the current study area [62]. 

13. Line 257-259, What is the pathway that infectious disease enhances DD? What are the infectious diseases you asked for?

We have mentioned about the pathway that infectious disease enhances DD as following:

In Discussion section:

While the children living in low socioeconomic status have high chances of occurrence of infectious diseases given the poor sanitation conditions [1], the occurrence of infectious disease in early years of life can lead to delayed development [63,64].

The infectious diseases we asked were medically reported Diarrhea, Malaria, Intestinal Parasite and specify if others. However, we received responses for Diarrhea, Intestinal Parasite and Acute Respiratory Infections only. We have included this information as following:

In Materials and Method section:

Similarly, biological variables included of birth weight (normal if ≥ 2500 gram, underweight < 2500 gram) and history of infectious diseases such as diarrhea, malaria, intestinal parasite, and others in the past six months (yes/no). However, the total number of medically reported histories of infectious diseases in the past six months by the respondents consisted of only Diarrhea, Intestinal Parasites and Acute Respiratory Infections (ARI). 

14. Above line 268

 What are the limitations of your study? This section is missing.

- small sample size, purposive sampling

- missing independent variables such as neighborhood effect and so on..

We have mentioned strengths and weaknesses of the study. 

In Discussion section:

Our study is one of those minimal studies that presents the developmental status of children living in urban slums in Nepal; one of the dimmed areas of developmental aspect for children from low-resource settings. The tool we used for the study assess the developmental status of a child in four developmental aspects such as gross motor, fine motor, language/ speech, and social development which are one of the major aspects considered primary for any of the developmental tools. However, we had few limitations such as; not having neighborhood and paternal characteristics incorporated within the questionnaire, having relatively small sample size. Also, there could have been a recall bias at times as there were few questions that would require respondents to report; if their child performed any of the activities, weight at birth and occurrence of infectious diseases within the past 6 months. Additionally, use of a cross-sectional study method limits the potential to examine the causal relationship.

15. In summary, you need to revise your writing using active voice, especially when you are talking about your work. And more importantly, you need to discuss more in your discussion section.

Thank you for your feedback. We have revised the manuscript and discussions using active voice as per the suggestions, where possible.

---

## [Decision Letter · Decision Letter 1]

13 Dec 2021

PONE-D-21-05500R1Developmental delay and its associated factors among children under five years in urban slums of NepalPLOS ONE

Dear Dr. Bishwokarma,

Thank you for submitting your manuscript to PLOS ONE. After careful consideration, we feel that it has merit but does not fully meet PLOS ONE’s publication criteria as it currently stands. Therefore, we invite you to submit a revised version of the manuscript that addresses the points raised during the review process.

Considering reviewers suggestion and my own reading of the paper, I am recommending a minor revision for this paper. Looking forward to read the revised paper.  Please submit your revised manuscript by Jan 27 2022 11:59PM. If you will need more time than this to complete your revisions, please reply to this message or contact the journal office at plosone@plos.org. Please include the following items when submitting your revised manuscript:A rebuttal letter that responds to each point raised by the academic editor and reviewer(s). You should upload this letter as a separate file labeled 'Response to Reviewers'.A marked-up copy of your manuscript that highlights changes made to the original version. You should upload this as a separate file labeled 'Revised Manuscript with Track Changes'.An unmarked version of your revised paper without tracked changes. You should upload this as a separate file labeled 'Manuscript'.If applicable, we recommend that you deposit your laboratory protocols in protocols.io to enhance the reproducibility of your results. Protocols.io assigns your protocol its own identifier (DOI) so that it can be cited independently in the future. For instructions see: https://journals.plos.org/plosone/s/submission-guidelines#loc-laboratory-protocols. Additionally, PLOS ONE offers an option for publishing peer-reviewed Lab Protocol articles, which describe protocols hosted on protocols.io. Read more information on sharing protocols at https://plos.org/protocols?utm_medium=editorial-email&utm_source=authorletters&utm_campaign=protocols.

We look forward to receiving your revised manuscript.

Kind regards,

Srinivas Goli, Ph.D.

Academic Editor

PLOS ONE

Journal Requirements:

Additional Editor Comments (if provided):

Considering reviewers suggestion and my own reading of the paper, I am recommending a minor revision for this paper. Looking forward to read the revised paper.

Reviewers' comments:

Reviewer's Responses to Questions

**Comments to the Author**

1. If the authors have adequately addressed your comments raised in a previous round of review and you feel that this manuscript is now acceptable for publication, you may indicate that here to bypass the “Comments to the Author” section, enter your conflict of interest statement in the “Confidential to Editor” section, and submit your "Accept" recommendation.

Reviewer #1: (No Response)

Reviewer #3: All comments have been addressed

2. Is the manuscript technically sound, and do the data support the conclusions?

Reviewer #1: Yes

Reviewer #3: Partly

3. Has the statistical analysis been performed appropriately and rigorously? 

Reviewer #1: Yes

Reviewer #3: No

4. Have the authors made all data underlying the findings in their manuscript fully available?

Reviewer #1: Yes

Reviewer #3: Yes

5. Is the manuscript presented in an intelligible fashion and written in standard English?

Reviewer #1: Yes

Reviewer #3: Yes

6. Review Comments to the Author

Reviewer #1: Dear Authors,

Thank you for your revision. There is one comment that is still of importance to reflect in your manuscript.

Please include the studies that have already done in Nepalese settings. Mentioning "Only few studies" will limit the significance/scientific merit of your manuscript. Briefly summaries those few studies related to developmental delays conducted in Nepal.

Thank you.

Reviewer #3: Dear authors,

Thank you so much for addressing my previous comments. Here are my comments regarding this revision.

1. Abstract: Your first and second sentence need some grammatical rephrasing. For example, you can omit unnecessary words like this. Children living in slum-like conditions of developing countries are at risk of exposure and threats that can adversely affect their ability to acquire full developmental potentials. I think you need to check this issue throughout your writing.

2. Methods: Is there any reason to select 30% of wards as states in "With that total number of wards remaining, 30% of the wards were selected by lottery method." or is it for convenience?

3. Line 159-161. Could you provide reference for your classification for privileged and underprivileged ethnicity classification? "Attributes of family ethnicity like Dalit, Disadvantaged Janajati, Non-Dalit Disadvantaged Terai Caste, and Religious Minorities were added to form Underprivileged and Relatively Advantaged Janajati and Upper Caste Group were added to form Privileged."

4. Line 191-192: " Additionally, to measure the economic situation of household and socio-economic status, we included IWI related questions within the questionnaire. Could you provide reference for this?

5. Line 206-208: While using logistic regression, how you decided your final model? Could you explain a bit in method section?

6. Table: 1 to 3: I think you should not mention minimum and maximum like that, you can provide range as: logistic regression: correlation among independent variables, what is your final model?

7. Line 247-255: While describing results, you can describe the association like this: On bivariate analysis, we found................" Otherwise, the meaning could be different.

8. Line 280-281: "Findings related to prevalence in our study is much higher as compared to the prevalence rate of other studies done in developed nations." If you are writing in past tense then, be consistent. "was" instead of "is". Please check it throughout the piece.

9. Line 311-314: "Though many studies have shown there’s a strong link between the low birth weight of children with their developmental status [58,59]. However, children in our study didn’t have the disadvantage of being born with low birth weight, which is consistent with study done among Chinese [7] and Brazilian children [60]." This sentence require grammatical correction.

10. Line 348-350: "Findings from the study suggest that there should be similar studies conducted among children living in slum-like conditions". How could you say this on the basis of your findings? Could you explain a bit?

Overall, the revision seems nice, but you need to revise your writing in terms of reducing the redundancy and improving the grammatical issues.

Best,

Reviewer

7. PLOS authors have the option to publish the peer review history of their article (what does this mean?). If published, this will include your full peer review and any attached files.

Reviewer #1: **Yes: **Mandira Adhikari, MScNg, MScGH (Research)

Reviewer #3: No

---

## [Author Response · Author response to Decision Letter 1]

9 Jan 2022

We are grateful to all the reviewers for their time and effort to read and comment on our paper. Their comments are constructive and the paper has benefitted from it. Below we present the point-by-point response to the reviewer’s comments. 

Comments to the Author

1. If the authors have adequately addressed your comments raised in a previous round of review and you feel that this manuscript is now acceptable for publication, you may indicate that here to bypass the “Comments to the Author” section, enter your conflict of interest statement in the “Confidential to Editor” section, and submit your "Accept" recommendation.

Reviewer #1: (No Response)

Reviewer #3: All comments have been addressed

Thank you for your feedback.

2. Is the manuscript technically sound, and do the data support the conclusions?

Reviewer #1: Yes

Reviewer #3: Partly

We have worked to make the manuscript technically sound and revised the conclusions based on the findings.

3. Has the statistical analysis been performed appropriately and rigorously?

Reviewer #1: Yes

Reviewer #3: No

Thank you for your feedback. We have performed statistical analysis under proper guidance from expertise. ________________________________________

4. Have the authors made all data underlying the findings in their manuscript fully available?

Reviewer #1: Yes

Reviewer #3: Yes

Thank you for the feedback.

5. Is the manuscript presented in an intelligible fashion and written in standard English?

Reviewer #1: Yes

Reviewer #3: Yes

Thank you for the feedback.

6. Review Comments to the Author

Reviewer #1: Dear Authors,

Thank you for your revision. There is one comment that is still of importance to reflect in your manuscript.

Please include the studies that have already done in Nepalese settings. Mentioning "Only few studies" will limit the significance/scientific merit of your manuscript. Briefly summaries those few studies related to developmental delays conducted in Nepal.

Thank you.

Thank you for pointing it out. We have added brief summary of relevant studies as per your suggestion.

In Introduction section:

Research estimating children at risk of development delay in overall aspects could be essential to highlight the need of interventions and policies targeting early childhood development [26]. Moreover, past studies have shown association between development outcomes and biological and anthropometrical [27], and nutritional factors [28] among children living in Nepal.

Reviewer #3: Dear authors,

Thank you so much for addressing my previous comments. Here are my comments regarding this revision.

1. Abstract: Your first and second sentence need some grammatical rephrasing. For example, you can omit unnecessary words like this. Children living in slum-like conditions of developing countries are at risk of exposure and threats that can adversely affect their ability to acquire full developmental potentials. I think you need to check this issue throughout your writing.

Thank you for your feedback. We have worked on grammar and rephrased first and second sentence. Also, we have worked on grammatical errors and language used throughout the writing as much as possible.

In Abstract section: 

Children from low-resource settings are more likely to encounter those factors that adversely influence their ability to acquire developmental potential. This study was conducted to assess the developmental status and its associated factors among children under five years of slum areas of Butwal Sub Metropolitan City, Rupandehi, Nepal. 

2. Methods: Is there any reason to select 30% of wards as states in "With that total number of wards remaining, 30% of the wards were selected by lottery method." or is it for convenience?

We selected 30% of the wards randomly for convenience as the study had to be conducted using limited resources. 

3. Line 159-161. Could you provide reference for your classification for privileged and underprivileged ethnicity classification? "Attributes of family ethnicity like Dalit, Disadvantaged Janajati, Non-Dalit Disadvantaged Terai Caste, and Religious Minorities were added to form Underprivileged and Relatively Advantaged Janajati and Upper Caste Group were added to form Privileged."

We have provided reference for the classification of privileged and underprivileged ethnicity as follows; however, we have also replaced it with Advantaged Group and Disadvantaged Group, respectively:

Attributes of family caste-ethnicity like Dalit, Disadvantaged Janajati, Non-Dalit Disadvantaged Terai Caste, and Religious Minorities were added to form Disadvantaged Group and Relatively Advantaged Janajati and Upper Caste Group were added to form Advantaged Group [31].

4. Line 191-192: " Additionally, to measure the economic situation of household and socio-economic status, we included IWI related questions within the questionnaire. Could you provide reference for this?

Thank you for your feedback. We have not provided any reference here because the International Wealth Index was already cited in the Study Parameters sub-section of Materials and Methods as follows. 

In Materials and Methods section:

Socio-demographic variables included of age in months and sex of children as stated by respondent, family ethnicity division (Dalit, Disadvantaged Janajati, Non-Dalit Disadvantaged, Terai Caste, Religious Minorities, Relatively Advantaged Janajati, Upper Caste Group) based on Health Management Information System of Nepal (HMIS) ethnicity classification [28], family type (nuclear, joint and extended), and socio-economic status based on International Wealth Index (IWI) classification [29] (extremely poor, poor, middle class, upper middle class, rich).

However, we have cited it again as your feedback. 

In Materials and Methods section:

Additionally, to measure the economic situation of household and socio-economic status, we included IWI related questions within the questionnaire [29].

5. Line 206-208: While using logistic regression, how you decided your final model? Could you explain a bit in method section?

Thank you for your feedback. We have provided explanation as follows. 

In Materials and Methods section:

Chi-square test and logistic regression were applied to determine the association between dependent and independent variables. All the statistical tests done were two-tailed and were considered statistically significant for a p-values<0.05 at 95% CI. While using logistics models, we adjusted for age of child, sex, family caste-ethnicity, family type, socio-economic status, mother’s age at birth of child, educational level, occupation, books available at home, language, parental smoking, parental alcoholism, birth weight, occurrence of infectious diseases in past 6 months, weight for height, weight for age and height for age.

6. Table: 1 to 3: I think you should not mention minimum and maximum like that, you can provide range as: logistic regression: correlation among independent variables, what is your final model?

Thank you for your suggestion. We have removed minimum and maximum values included in the table.

7. Line 247-255: While describing results, you can describe the association like this: On bivariate analysis, we found................" Otherwise, the meaning could be different.

On bivariate analysis, the age of children (p = 0.003) and their socio-economic status (p = 0.049) were associated with their developmental status. Status of availability of learning materials for children was associated (p <0.001) with the developmental status.

8. Line 280-281: "Findings related to prevalence in our study is much higher as compared to the prevalence rate of other studies done in developed nations." If you are writing in past tense then, be consistent. "was" instead of "is". Please check it throughout the piece.

Thank you for pointing it out. We have revised it as follows:

In Discussion section:

Findings related to prevalence in our study was much higher as compared to the prevalence rate of other studies done in developed nations [38,39]. 

9. Line 311-314: "Though many studies have shown there’s a strong link between the low birth weight of children with their developmental status [58,59]. However, children in our study didn’t have the disadvantage of being born with low birth weight, which is consistent with study done among Chinese [7] and Brazilian children [60]." This sentence require grammatical correction.

Thank you for your feedback. We have revised the sentence as per your suggestion.

In Discussion section:

Though many studies have shown there’s a strong link between the low birth weight of children with their developmental status [58,59]. However, children in our study had no consequence of being low weight at birth to their development, which is similar to the findings from the study done among Chinese [7] and Brazilian children [60]. 

10. Line 348-350: "Findings from the study suggest that there should be similar studies conducted among children living in slum-like conditions". How could you say this on the basis of your findings? Could you explain a bit?

Thank you for pointing it out. We included this suggestion based on the finding about developmental delay which is high among children living in slum-like conditions in the present study. However, we have made some changes in the sentence as we wanted to indicate that interventions and research are needed to address such issues. 

In Conclusions section: 

Findings from the study suggest that investigations need to focus on overall developmental aspects of early childhood development of children.

Overall, the revision seems nice, but you need to revise your writing in terms of reducing the redundancy and improving the grammatical issues.

Thank you for your feedback. We have tried worked on reducing the redundancy and improving the grammatical errors.

---

## [Editor Report · Decision Letter 2]

13 Jan 2022

Developmental delay and its associated factors among children under five years in urban slums of Nepal

PONE-D-21-05500R2

Dear Dr. Bishwokarma,

We’re pleased to inform you that your manuscript has been judged scientifically suitable for publication and will be formally accepted for publication once it meets all outstanding technical requirements.

Kind regards,

Srinivas Goli, Ph.D.

Academic Editor

PLOS ONE

Additional Editor Comments (optional):

Recommending this piece for publication in PLOS One. Congratulations to authors.
---

## [Editor Report · Acceptance letter]

2 Feb 2022

PONE-D-21-05500R2 

Developmental delay and its associated factors among children under five years in urban slums of Nepal 

Dear Dr. Bishwokarma:

I'm pleased to inform you that your manuscript has been deemed suitable for publication in PLOS ONE. Congratulations! Your manuscript is now with our production department. 

Kind regards, 

on behalf of

Dr. Srinivas Goli 

Academic Editor

PLOS ONE